# GENERATIVE RATIO MATCHING NETWORKS

**Akash Srivastava**[*]
MIT-IBM Watson AI Lab[†]
akash.srivastava@ibm.com

**Kai Xu**[*]
University of Edinburgh
kai.xu@ed.ac.uk

**Michael U. Gutmann**
University of Edinburgh
michael.gutmann@ed.ac.uk

**Charles Sutton**
Google AI[‡]
charlessutton@google.com

## ABSTRACT

Deep generative models can learn to generate realistic-looking images, but many of the most effective methods are adversarial and involve a saddlepoint optimization, which requires a careful balancing of training between a generator network and a critic network. Maximum mean discrepancy networks (MMD-nets) avoid this issue by using kernel as a fixed adversary, but unfortunately, they have not on their own been able to match the generative quality of adversarial training. In this work, we take their insight of using kernels as fixed adversaries further and present a novel method for training deep generative models that does not involve saddlepoint optimization. We call our method **g**enerative **ra**tio **m**atching or GRAM for short. In GRAM, the generator and the critic networks do not play a zero-sum game against each other, instead they do so against a fixed kernel. Thus GRAM networks are not only stable to train like MMD-nets but they also match and beat the generative quality of adversarially trained generative networks. [1]

## 1 INTRODUCTION

Deep generative models (Kingma & Welling, 2013; Goodfellow et al., 2014; Kingma & Dhariwal, 2018) have been shown to learn to generate realistic-looking images. These methods train a deep neural network, called a generator, to transform samples from a noise distribution to samples from the data distribution. Most methods use adversarial learning (Goodfellow et al., 2014), in which the generator is pitted against a critic function, also called a discriminator, which is trained to distinguish between the samples from the data distribution and from the generator. Upon successful training the two sets of samples become indistinguishable with respect to the critic.

Maximum mean discrepancy networks (MMD-nets) (Li et al., 2015; Dziugaite et al., 2015) are a class of generative models that are trained to minimize the MMD (Gretton et al., 2012) between the true data distribution and the model distribution. MMD-nets are similar in spirit to generative adversarial networks (GANs) (Goodfellow et al., 2014; Nowozin et al., 2016), in the sense that the MMD is defined by maximizing over a class of critic functions. However, in contrast to GANs, where finding the right balance between generator and critic is difficult, training is simpler for MMD-nets because using the kernel trick the MMD can be estimated without the need to numerically optimize over the critic function. This avoids the need in GANs to numerically solve a saddlepoint problem.

Unfortunately, although MMD-nets work well on low dimensional data, these networks have not on their own matched the generative performance of adversarial methods on higher dimensional datasets, such as natural images (Dziugaite et al., 2015). Several authors (Li et al., 2017; Bińkowski et al., 2018a) suggest that a reason is that MMD is sensitive to the choice of kernel. Li et al. (2017) propose a method called MMD-GAN, in which the critic maps the samples from the generator and the data into a lower-dimensional representation, and MMD is applied in this transformed space. This

---

[*]Equal contributions.
[†]Additional affiliations: The University of Edinburgh and IBM Research.
[‡]Additional affiliations: The University of Edinburgh and The Alan Turing Institute

[1]Official implementations are available at https://github.com/GRAM-nets.

can be interpreted as a method for learning the kernel in MMD. The critic is learned adversarially by maximizing the MMD at the same time as it is minimized with respect to the generator. This is much more effective than MMD-nets, but training MMD-GANs can be challenging, because in this saddlepoint optimization problem, the need to balance training of the learned kernel and the generator can create a sensitivity to hyperparameter settings. In practice, it is necessary to introduce several additional penalties to the loss function in order for training to be effective.

In this work, we present a novel training method that builds on MMD-nets' insight to use kernels as fixed adversaries in order to avoid saddlepoint optimization based training for the critic and the generator. Our goal is for the critic to map the samples into a lower-dimensional space in which the MMD-net estimator will be more effective. Our proposal is that the critic should be trained to preserve density ratios, namely, the ratio of the true density to the model density. If the critic is successful in this, then matching the generator to the true data in the lower dimensional space will also match the distributions in the original space. We call networks that have been trained using this criterion *generative ratio matching (GRAM)* networks, or GRAM-nets[2]. We show empirically that our method is not only able to generate high quality images but by virtue of avoiding a zero-sum game (in critic and generator) it avoids saddlepoint optimization and hence is more stable to train and robust to the choice of hyperparameters.

## 1.1 RELATED WORK

Li et al. (2015) and Dziugaite et al. (2015) independently proposed MMD-nets, which use the MMD criterion to train a deep generative model. Unlike $f$-divergences, MMD is well defined even for distributions that do not have overlapping support, which is an important consideration for training generative models (Arjovsky et al., 2017). Therefore, MMD-nets use equation (2) in order to minimize the discrepancy between the distributions $q_x$ and $p_x$ with respect to $G_\gamma$. However, the sample quality of MMD-nets generally degrades for higher dimensional or color image datasets (Li et al., 2015).

To address this problem, Li et al. (2017) introduce MMD-GANs, which use a critic $f_\theta : \mathbb{R}^D \mapsto \mathbb{R}^K$ to map the samples to a lower dimensional space $\mathbb{R}^K$, and train the generator to minimize MMD in this reduced space. This can be interpreted as learning the kernel function for MMD, because if $f_\theta$ is injective and $k_0$ is a kernel in $\mathbb{R}^K$, then $k(x, x') = k_0(f_\theta(x), f_\theta(x'))$ is a kernel in $\mathbb{R}^D$. This injectivity constraint on $f_\theta$ is imposed by introducing another deep neural network $f'_\phi$, which is trained to approximately invert $f_\theta$ using an auto-encoding penalty. Though it has been shown before that inverting $f_\theta$ is not necessary for the method to work. We also confirm this in experiments (See appendix).

The critic $f_\theta$ is trained using an adversarial criterion (maximizing equation (2) that the generator minimizes), which requires numerical saddlepoint optimization, and avoiding this was one of the main attractions of MMD in the first place. Due to this, successfully training $f_\theta$ in practice required a penalty term called feasible set reduction on the class of functions that $f_\theta$ can learn to represent. Furthermore, $f$ is restricted to be $k$-Lipschitz continuous by using a low learning rate and explicitly clipping the gradients during update steps of $f$ akin to WGAN (Arjovsky et al., 2017). Recently, Bińkowski et al. (2018b); Li et al. (2019) have proposed improvements to stabilize the training of the MMD-GAN method. But these method still rely on solving the same saddle-point problem to train the critic.

## 2 BACKGROUND

Given data $x_i \in \mathbb{R}^D$ for $i \in \{1 \dots N\}$ from a distribution of interest with density $p_x$, the goal of deep generative modeling is to learn a parameterized function $G_\gamma : \mathbb{R}^h \mapsto \mathbb{R}^D$, called a generator network, that maps samples $z \in \mathbb{R}^h$ where $h < D$ from a noise distribution $p_z$ to samples from the model distribution. Since $G_\gamma$ defines a new random variable, we denote its density function by $q_x$, and also write $x^q = G_\gamma(z)$, where we suppress the dependency of $x^q$ on $\gamma$. The parameters $\gamma$ of the generator are chosen to minimize a loss criterion which encourages $q_x$ to match $p_x$.

---

[2]Interestingly, the training of GRAM-nets heavily relies on the use of kernel Gram matrices.

## 2.1 MAXIMUM MEAN DISCREPANCY

Maximum mean discrepancy measures the discrepancy between two distributions as the maximum difference between the expectations of a class of functions $\mathcal{F}$, that is,

$$\text{MMD}_{\mathcal{F}}(p, q) = \sup_{f \in \mathcal{F}} \left( \mathbb{E}_p[f(x)] - \mathbb{E}_q[f(x)] \right), \tag{1}$$

where $\mathbb{E}$ denotes expectation. If $\mathcal{F}$ is chosen to be a rich enough class, then $\text{MMD}(p, q) = 0$ implies that $p = q$. Gretton et al. (2012) show that it is sufficient to choose $\mathcal{F}$ to be a unit ball in an reproducing kernel Hilbert space $\mathcal{R}$ with a characteristic kernel $k$. Given samples $x_1 \ldots x_N \sim p$ and $y_i \ldots y_M \sim q$, we can estimate $\text{MMD}_{\mathcal{R}}$ as

$$\hat{\text{MMD}}_{\mathcal{R}}^2(p, q) = \frac{1}{N^2} \sum_{i=1}^{N} \sum_{i'=1}^{N} k(x_i, x_{i'}) - \frac{2}{NM} \sum_{i=1}^{N} \sum_{j=1}^{M} k(x_i, y_j) + \frac{1}{M^2} \sum_{j=1}^{M} \sum_{j'=1}^{M} k(y_j, y_{j'}). \tag{2}$$

## 2.2 DENSITY RATIO ESTIMATION

Sugiyama et al. (2012) present an elegant MMD-based estimator for the ratio between the densities $p$ and $q$; $r(x) = \frac{p(x)}{q(x)}$ that only needs samples from $p$ and $q$. This estimator is derived by optimizing

$$\min_{r \in \mathcal{R}} \left\| \int k(x; .)p(x)dx - \int k(x; .)r(x)q(x)dx \right\|_{\mathcal{R}}^2, \tag{3}$$

where $k$ is a kernel function. It is easy to see that at the minimum, we have $r = p/q$. This estimator is not guaranteed to be non-negative. As such, a positivity constraint should be imposed if needed.

Sugiyama et al. (2011) suggest that density ratio estimation for distributions $p$ and $q$ over $\mathbb{R}^D$ can be more accurately done in lower dimensional sub-spaces $\mathbb{R}^K$. They propose to first learn a linear projection to a lower dimensional space by maximizing an $f$-divergence between the distributions $\bar{p}$ and $\bar{q}$ of the projected data and then estimate the ratio of $\bar{p}$ and $\bar{q}$ (using direct density ratio estimation). They showed that the projected distributions preserve the original density ratio.

## 3 METHOD

Our aim will be to enjoy the advantages of MMD-nets, but to improve their sample quality by mapping the data ($\mathbb{R}^D$) into a lower-dimensional space ($\mathbb{R}^K$), using a critic network $f_\theta$, before computing the MMD criterion. Because MMD with a fixed kernel performs well for lower-dimensional data (Li et al., 2015; Dziugaite et al., 2015), we hope that by choosing $K < D$, we will improve the performance of the MMD-net. Instead of training $f_\theta$ using an adversarial criterion like MMD-GAN, we aim at a stable training method that avoids the saddle-point optimization for training the critic.

More specifically, unlike the MMD-GAN type methods, instead of maximizing the same MMD criterion that the generator is trained to minimize, we train $f_\theta$ to minimize the *squared ratio difference*, that is, the difference between density ratios in the original space and in the low-dimensional space induced by $f_\theta$ (Section 3.1). More specifically, let $\bar{q}$ be the density of the transformed simulated data, i.e., the density of the random variable $f_\theta(G_\gamma(z))$, where $z \sim p_z$. Similarly let $\bar{p}$ be the density of the transformed data, i.e., the density of the random variable $f_\theta(x)$. The squared ratio difference is minimized when $\theta$ is such that $p_x/q_x$ equals $\bar{p}/\bar{q}$. The motivation is that if density ratios are preserved by $f_\theta$, then matching the generator to the data in the transformed space will also match it in the original space (Section 3.2). The reduced dimension of $f_\theta$ should be chosen to strike a trade-off between dimensionality reduction and ability to approximate the ratio. If the data lie on a lower-dimensional manifold in $\mathbb{R}^D$, which is the case for e.g. natural images, then it is reasonable to suppose that we can find a critic that strikes a good trade-off. To compute this criterion, we need to estimate density ratio $\bar{p}/\bar{q}$, which can be done in closed form using MMD (Section 2.2). The generator is trained as an MMD-net to match the transformed data $\{f_\theta(x_i)\}$ with transformed outputs from the generator $\{f(G_\gamma(z_i)\}$ in the lower dimensional space (Section 3.2). Our method performs stochastic gradient (SG) optimization on the critic and the generator jointly (Section 3.3).

### 3.1 Training the Critic using Squared Ratio Difference

Our principle is to choose $f_\theta$ so that the resulting densities $\bar{p}$ and $\bar{q}$ preserve the density ratio between $p_x$ and $q_x$. We will choose $f_\theta$ to minimize the distance between the two density ratio functions

$$r_x(x) = p_x(x)/q_x(x) \qquad r_\theta(x) = \bar{p}(f_\theta(x))/\bar{q}(f_\theta(x)).$$

One way to measure how well $f_\theta$ preserves density ratios is to use the squared distance

$$D^*(\theta) = \int q_x(x) \left( \frac{p_x(x)}{q_x(x)} - \frac{\bar{p}(f_\theta(x))}{\bar{q}(f_\theta(x))} \right)^2 dx. \tag{4}$$

This objective is minimized only when the ratios match exactly, that is, when $r_x = r_\theta$ for all $x$ in the support of $q_x$. Clearly a distance of zero can be trivially achieved if $K = D$ and if $f_\theta$ is the identity function.But nontrivial optima can exist as well. For example, suppose that $p_x$ and $q_x$ are "intrinsically low dimensional" in the following sense. Suppose $K < D$, and consider two distributions $p_0$ and $q_0$ over $\mathbb{R}^K$, and an injective map $T : \mathbb{R}^K \to \mathbb{R}^D$. Suppose that $T$ maps samples from $p_0$ and $q_0$ to samples from $p_x$ and $q_x$, by which we mean $p_x(x) = J(\mathbf{D}T)p_0(T^{-1}(x))$, and similarly for $q_x$. Here $J(\mathbf{D}T)$ denotes the Jacobian $J(\mathbf{D}T) = \sqrt{|\delta T \delta T^\top|}$ of $T$. Then we have that $D^*$ is minimized to 0 when $f_\theta = T^{-1}$.

However, it is difficult to optimize $D^*(\theta)$ directly because density ratio estimation in high dimension, where data lives, is known to be hard, i.e. the term $\frac{p_x(x)}{q_x(x)}$ in (4) is difficult to estimate. We will show how to alleviate this issue next.

**Avoiding density ratio estimation in data space:** To avoid computing the term $\frac{p_x(x)}{q_x(x)}$ in (4), we expand the square in (4), apply the law of the unconscious statistician and cancel terms out (See appendix A for detailed steps), which yields

$$
\begin{aligned}
D^*(\theta) &= C + \int \bar{q}(f_\theta(x)) \left( \frac{\bar{p}(f_\theta(x))}{\bar{q}(f_\theta(x))} \right)^2 df_\theta(x) - 2 \int \bar{p}(f_\theta(x)) \frac{\bar{p}(f_\theta(x))}{\bar{q}(f_\theta(x))} df_\theta(x), \\
&= C' - \left[ \int \bar{q}(f_\theta(x)) \left( \frac{\bar{p}(f_\theta(x))}{\bar{q}(f_\theta(x))} \right)^2 df_\theta(x) - 1 \right]
\end{aligned}
\tag{5}
$$

where $C$ and $C' = C - 1$ does not depend on $\theta$. This implies that minimizing $D^*$ is equivalent to maximizing the Pearson divergence

$$\text{PD}(\bar{p}, \bar{q}) = \int \bar{q}(f_\theta(x)) \left( \frac{\bar{p}(f_\theta(x))}{\bar{q}(f_\theta(x))} \right)^2 df_\theta(x) - 1 = \int \bar{q}(f_\theta(x)) \left( \frac{\bar{p}(f_\theta(x))}{\bar{q}(f_\theta(x))} - 1 \right)^2 df_\theta(x) \tag{6}$$

between $\bar{p}$ and $\bar{q}$, which justifies our terminology of referring to $f_\theta$ as a critic function. So we can alternatively interpret our squared ratio distance objective as preferring $f_\theta$ so that the low-dimensional distributions $\bar{p}$ and $\bar{q}$ are maximally separated under Pearson divergence. Therefore $D^*$ can be minimized empirically using samples $x_1^q \ldots x_M^q \sim q_x$, to maximize the critic loss function

$$\mathcal{L}(\theta) = \frac{1}{M} \sum_{i=1}^{M} \left[ r_\theta(x_i^q) - 1 \right]^2. \tag{7}$$

Optimizing this requires a way to estimate $r_\theta(x_i^q)$. For this purpose we use the density ratio estimator introduced in Section 2.2. Notice that to compute (7), we need the value of $r_\theta$ *only for* the points $x_1^q \ldots x_M^q$. In other words, we need to approximate the vector $\mathbf{r}_{q,\theta} = [r_\theta(x_1^q) \ldots r_\theta(x_M^q)]^T$. Following Sugiyama et al. (2012), we replace the integrals in (3) with Monte Carlo averages over the points $f_\theta(x_1^q) \ldots f_\theta(x_M^q)$ and over points $f_\theta(x_1^p) \ldots f_\theta(x_N^p) \sim \bar{p}$; the minimizing values of $\mathbf{r}_{q,\theta}$ can then be computed as

$$\hat{\mathbf{r}}_{q,\theta} = \mathbf{K}_{q,q}^{-1} \mathbf{K}_{q,p} \mathbf{1}. \tag{8}$$

Here $\mathbf{K}_{q,q}$ and $\mathbf{K}_{q,p}$ denote the Gram matrices defined by $[\mathbf{K}_{q,q}]_{i,j} = k(f_\theta(x_i^q), f_\theta(x_j^q))$ and $[\mathbf{K}_{q,p}]_{i,j} = k(f_\theta(x_i^q), f_\theta(x_j^p))$.

Substituting these estimates into (7) and adding a positivity constraint $\hat{\mathbf{r}}_{q,\theta}^T.\mathbf{1}$ for using the MMD density ratio estimator (Sugiyama et al., 2012), we get

$$\hat{\mathcal{L}}(\theta) = \frac{1}{M} \sum_{i=1}^{M} [r_\theta(x_i^q) - 1]^2 + \lambda \hat{\mathbf{r}}_{q,\theta}^T.\mathbf{1}, \tag{9}$$

where $\lambda$ is a parameter to control the constraint, being set to 1 in all our experiments.[3] This objective can be maximised to learn the critic $f_\theta$. We see that this is an estimator of the Pearson divergence $\mathrm{PD}(\bar{p}, \bar{q})$ in that we are both, averaging over samples from $q_x$, and approximating the density ratio. Thus maximising this objective leads to the preservation of density ratio (Sugiyama et al., 2011).

## 3.2 Generator Loss

To train the generator network $G_\gamma$, we minimize the MMD in the low-dimensional space, where both the generated data and the true data are transformed by $f_\theta$. In other words, we minimize the MMD between $\bar{p}$ and $\bar{q}$. We sample points $z_1 \ldots z_M \sim p_z$ from the input distribution of the generator. Then using the empirical estimate (2) of the MMD, we define the generator loss function as

$$\hat{\mathcal{L}}^2(\gamma) = \frac{1}{N^2} \sum_{i=1}^{N} \sum_{i'=1}^{N} k(f_\theta(x_i), f_\theta(x_{i'})) - \frac{2}{NM} \sum_{i=1}^{N} \sum_{j=1}^{M} k(f_\theta(x_i), f_\theta(G_\gamma(z_j)))$$
$$+ \frac{1}{M^2} \sum_{j=1}^{M} \sum_{j'=1}^{M} k(f_\theta(G_\gamma(z_j)), f_\theta(G_\gamma(z_{j'}))) \tag{10}$$

which we minimize with respect to $\gamma$.

## 3.3 The Generative Ratio Matching Algorithm

Finally, the overall training of the critic and the generator proceeds by jointly performing SG optimization on $\hat{\mathcal{L}}(\theta)$ and $\hat{\mathcal{L}}(\gamma)$. Unlike WGAN (Arjovsky et al., 2017) and MMD-GAN (Li et al., 2017), we do not require the use of gradient clipping, feasible set reduction and autoencoding regularization terms. Our algorithm is a simple iterative process.

**while** *not converged* **do**
> Sample a minibatch of data $\{x_i^p\}_{i=1}^{N} \sim p_x$ and generated samples $\{x_i^q\}_{i=1}^{M} \sim q_x$;
> Using $f_\theta$ to transform data as $\{f_\theta(x_i^p)\}_{i=1}^{N}$ and generated samples as $\{f_\theta(x_i^q)\}_{i=1}^{M}$;
> Compute the Gram matrix $\mathbf{K}$ under Gaussian kernels in the transformed space;
> Compute $\hat{\mathcal{L}}(\theta)$ via (8) and (9), and $\hat{\mathcal{L}}(\gamma)$ via (10) using the same $\mathbf{K}$;
> Compute the gradients $g_\theta = \nabla_\theta \hat{\mathcal{L}}(\theta)$ and $g_\gamma = \nabla_\gamma \hat{\mathcal{L}}(\gamma)$ ;
> $\theta \leftarrow \theta + \eta g_\theta; \gamma \leftarrow \gamma - \eta g_\gamma$ ;          // Perform SG optimisation for $\theta$ and $\gamma$

**end**

**Algorithm 1:** Generative ratio matching

**Convergence:**    If we succeed in matching the generator to the true data in the low-dimensional space, then we have also matched the generator to the data in the original space, in the limit of infinite data. To see this, suppose that we have $\gamma^*$ and $\theta^*$ such that $D^*(\theta^*) = 0$ and that $M_y = \mathrm{MMD}(\bar{p}, \bar{q}) = 0$. Then for all $x$, we have $r_x(x) = r_{\theta^*}(x)$ because $D^*(\theta^*) = 0$, and that $r_{\theta^*}(x) = 1$, because $M_y = 0$. This means that $r_x(x) = 1$, so we have that $p_x = q_x$.

## 3.4 Stability of GRAM-nets

Unlike GANs, the GRAM formulation avoids the saddle-point optimization which leads to a very stable training of the model. In this section we provide a battery of controlled experiments to

---

[3]As it is pointed out by one of the reviewer, instead of adding the regularization term, another way to resolve the non-negativity issue of the ratio estimator is to simply clipping it the estimator be positive. This in fact works well in practice and can further improves the stability of training.

empirically demonstrate that GRAM training relying on the empirical estimators of MMD and Pearson Divergence (PD), i.e. equations (7) and (9). While the MMD estimator is very extensively studied, our novel empirical estimator of PD is not. Therefore we now show that in fact maximizing our estimator for PD can be reliably used for training the critic, i.e. it minimizes equation (4).

Stability of adversarial methods has been extensively studied before by using a simple dataset of eight axis-aligned Gaussian distributions with standard deviations of $0.01$, arranged in a ring shape (Roth et al., 2017; Mescheder et al., 2017; Srivastava et al., 2017). Therefore we train a simple generator using GRAM on this 2D ring dataset to study the stability and accuracy of GRAM-nets. We set the projection dimension $K = D = 2$ in order to facilitate visualization. Both the generator and the projection networks are implemented as two-layer feed-forward neural networks with two hidden layers of size of 100 and ReLU activations.

The generator output (orange) is visualized over the real data (blue) in Figure 1a at 10,100,1000 and 10,000th training iterations. The top row visualizes the observation space ($\mathbb{R}^D$) and the bottom row visualizes the projected space ($\mathbb{R}^K$). Note, as the training progress the critic (bottom row) tries to find projections that better separate the data and the generator distributions (especially noticeable in columns 1 and 3). This provides a clear and strong gradient signal to the generator optimization that successfully learns to minimize the MMD of the projected distributions and eventually the data and the model distributions as well (as shown in the last column). Notice, how in the final column the critic is no longer able to separate the distributions. Throughout this process the critic and the generator objectives are consistently minimized. This is clearly shown in Figure 1b which records the values of equations (10) for the generator (orange) loss and equation (5) for the critic objective (blue).

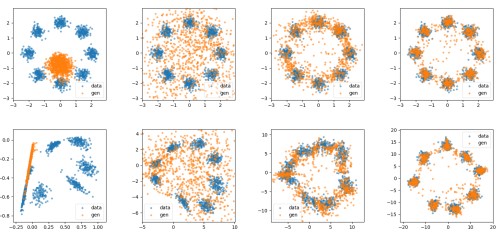 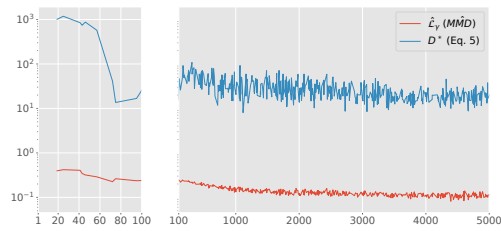

(a) Data and samples in the original (top) and projected space (bottom) during training; four plots are at iteration 10, 100, 1000 and 10,000 respectively. Notice how the projected space separates $\bar{p}$ and $\bar{q}$.

(b) Trace of $\hat{\mathcal{L}}_\gamma$ and $D^*$ (equation (5)) during training. The left plot is for iteration 1 to 100 and the right plot is for 100 to 5,000, with the same y-axes in the log scale.

Figure 1: Training results with projected dimension fixed to 2.

Next, we compare GRAM-based training against classical adversarial training that involves a zero-sum game. For this purpose we train a GAN with the same exact architecture and hyperparameter as those used for the GRAM-net in the previous experiments. Adversarial models are known to be unstable on the 2D ring dataset (Roth et al., 2017; Mescheder et al., 2017) as the critic can often easily outperform the generator in this simple setting. Figure 2 summarizes the results of this comparison. Both models are trained on three different levels of generator capacity and four different dimensions of the input noise variable ($h$). GANs are known to be unstable at both low and high dimensional noise but they additionally tend to mode-collapse (Srivastava et al., 2017) on high dimensional noise. This is confirmed in our experiments; GANs failed to learn the true data distribution in every case and in most cases the optimization also diverged. In sharp contrast to this, GRAM training successfully converged in all 12 settings. Adversarial training needs a careful balance between the generator and the critic capacities. In the plot we can see that, as the capacity of the generator becomes larger, the training become more unstable in GAN. This is not the case for GRAM-nets, which train well without requiring to make any adjustments in other hyperparameters. This enlists as an important advantage of GRAM, as one can use the most powerful model (generator and critic) given their computational budget instead of worrying about balancing the optimization. We also provide the corresponding results for MMD-nets and MMD-GANs in Appendix B.3, in which one can see MMD-nets can also capture all modes, but the learned distribution is less separated (or sharp) compared to GRAM-nets and MMD-GANs tend to produces distributions which are too sharp.

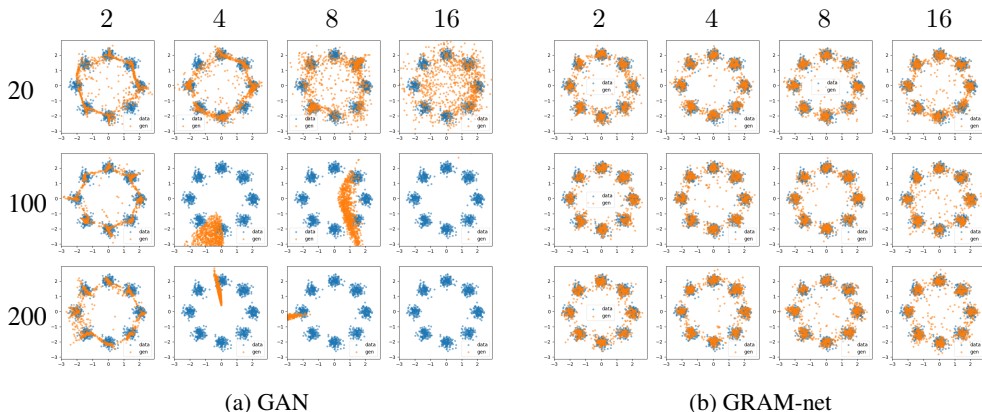

(a) GAN                                                                        (b) GRAM-net

Figure 2: Training after 2,000 epochs by varying noise dimension $h$ and the hidden layer size of critic model. For each model, each row is a different layer size in $[20, 100, 200]$ and each column is a different $h$ in $[2, 4, 8, 16]$. Half of the GAN training diverges while all GRAM training converges.

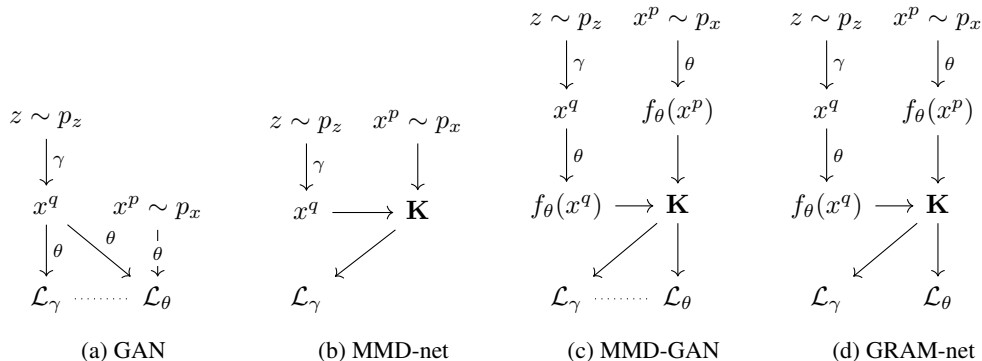

(a) GAN                    (b) MMD-net                    (c) MMD-GAN                    (d) GRAM-net

Figure 3: Computation graphs of GAN, MMD-net, MMD-GAN and GRAM-net. $\mathbf{K}$ is the kernel Gram matrix. Solid arrows represents the flow of the computation and dashed lines represents min-max relationship between the losses, i.e. saddle-point optimization in which minimizing one loss maximizes the other. Therefore in the zero-sum game case (GAN, MMD-GAN) the two objectives ($L_\gamma$ and $L_\theta$) cannot be optimized simultaneously (Mescheder et al., 2017).

### 3.4.1 COMPUTATION GRAPHS

Figure 3 visualizes the computational graph for GAN, MMD-net, MMD-GAN and GRAM-net. Solid arrows describe the direction of the computation, $\mathbf{K}$ denotes the kernel gram matrix and most importantly dashed lines represent the presence of saddlepoint optimization. In case of GAN and MMD-GAN, the dashed lines imply that by formulation, training the critic adversarially affects the training of the generator as they are trained by minimizing and maximizing the same objective i.e. $L_\gamma = -L_\theta$. Both MMD-nets and GRAM-nets avoid this saddle-point problem. In GRAM-nets, the critic and generator do not play a zero-sum game as they are trained to optimize different objectives, i.e. the critic learns to find a projection in which the density ratios of the pair of input distributions are preserved after the projection and the generator tries to minimize the MMD in the projected space.

## 4 EXPERIMENTS

In this section we empirically compare GRAM-nets against MMD-GANs and vanilla GANs, on the Cifar10 and CelebA image datasets. Please note that while we have tried include maximum number of evaluations in this section itself, due to space limitations, some of the results are made available in the appendix. To evaluate the sample quality and resilience against mode dropping, we used Frechet

Inception Distance (FID) (Heusel et al., 2017).[4] Like the Inception Score (IS), FID also leverages a pre-trained Inception Net to quantify the quality of the generated samples, but it is more robust to noise than IS and can also detect intra-class mode dropping (Lucic et al., 2017). FID first embeds both the real and the generated samples into the feature space of a specific layer of the pre-trained Inception Net. It further assumes this feature space to follow a multivariate Gaussian distribution and calculates the mean and covariance for both sets of embeddings. The sample quality is then defined as the Frechet distance between the two Gaussian distributions, which is

$$\text{FID}(x_p, x_q) = \|\mu_{x_p} - \mu_{x_q}\|_2^2 + \text{Tr}(\Sigma_{x_p} + \Sigma_{x_q} - 2(\Sigma_{x_p}\Sigma_{x_q})^{\frac{1}{2}}),$$

where $(\mu_{x_p}, \Sigma_{x_p})$, and $(\mu_{x_q}, \Sigma_{x_q})$ are the mean and covariance of the sample embeddings from the data distribution and model distribution. We report FID on a held-out set that was not used to train the models. We run all the models three times from random initializations and report the mean and standard deviation of FID over the initializations.

*Architecture:* We test all the methods on the same architectures for the generator and the critic, namely a four-layer DCGAN architecture (Radford et al., 2015), because this has been consistently shown to perform well for the datasets that we use. Additionally, to study the effect of changing architecture, we also evaluate a slightly weaker critic, with the same number of layers but half the number of hidden units. Details of the architectures are provided in Appendix D.

*Hyperparameters:* To facilitate fair comparison with MMD-GAN we set all the hyperparameters shared across the three methods to the values used in Li et al. (2017). Therefore, we use a learning rate of $5e^{-5}$ and set the batch size to $64$. For the MMD-GAN and GRAM-nets, we used the same set of RBF kernels that were used in Li et al. (2017). We used the implementation of MMD-GANs from Li et al. (2017).[5] We leave all the hyper-parameters that are only used by MMD-GAN to the settings in the authors' original code. For GRAM-nets, we choose $K = h$, i.e. the projected dimension equals the input noise dimension. We present an evaluation of hyperparameter sensitivity in Section 4.2.

Table 1: Sample quality (measured by FID; lower is better) of GRAM-nets compared to GANs.

| Arch. | Dataset | MMD-GAN | GAN | GRAM-net |
|---|---|---|---|---|
| **DCGAN** | Cifar10 | $40.00 \pm 0.56$ | $26.82 \pm 0.49$ | **$24.85 \pm 0.94$** |
| **Weaker** | Cifar10 | $210.85 \pm 8.92$ | $31.64 \pm 2.10$ | **$24.82 \pm 0.62$** |
| **DCGAN** | CelebA | $41.105 \pm 1.42$ | $30.97 \pm 5.32$ | **$27.04 \pm 4.24$** |

Table 2: FID with fully convolutional architecture originally used by Li et al. (2017).

| Dataset | MMD-GAN |
|---|---|
| **Cifar10** | $38.39 \pm 0.28$ |
| **CelebA** | $40.27 \pm 1.32$ |

## 4.1 IMAGE QUALITY

We now look at how our method competes against GANs and MMD-GANs on sample quality and mode dropping on Cifar10 and CelebA datasets. Results are shown in Table 1. Clearly, GRAM-nets outperform both baselines. For CelebA, we do not run experiments using the weaker critic, because this is a much larger and higher-dimensional dataset, so a low-capacity critic is unlikely to work well.

It is worth noting that while the difference between the FIDs of GAN and GRAM-net is relatively smaller, it is quite significant that GRAM-net outperforms GAN on both datasets. As shown in a large scale study of adversarial generative models (Lucic et al., 2017), GANs in general perform very well on FID when compared to the state-of-the-art methods such as WGAN (Arjovsky et al., 2017). Interestingly, in the case CIFAR10, GANs are the state-of-art on FID performance.

To provide evidence that GRAM-nets are not simply memorizing the training set, we note that we measure FID on a held-out set, so a network that memorized the training set would be likely to have poor performance. For additional qualitative evidence of this, see Figure 4. This figure shows the five nearest neighbors from the training set for 20 randomly generated samples from the trained generator of our GRAM-net. None of the generated images have an exact copy in the training set, and qualitatively the 20 images appear to be fairly diverse.

Note that our architecture is different from that used in the results of Li et al. (2017). That work uses a fully convolutional architecture for both the critic and the generator, which results in an

---

[4]We use a standard implementation available from `https://github.com/bioinf-jku/TTUR`
[5]Available at `https://github.com/OctoberChang/MMD-GAN`.

Figure 4: Nearest training images to samples from a GRAM-net trained on Cifar10. In each column, the top image is a sample from the generator, and the images below it are the nearest neighbors.

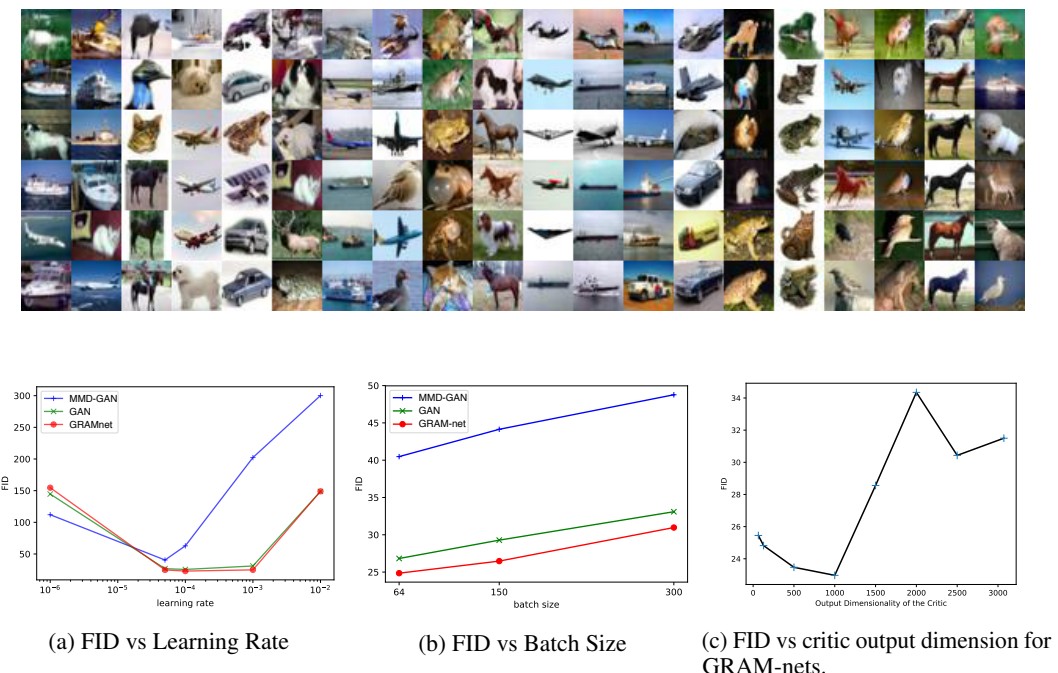

(a) FID vs Learning Rate

(b) FID vs Batch Size

(c) FID vs critic output dimension for GRAM-nets.

Figure 5: Hyper-parameter sensitivity of MMD-GAN, GAN and GRAM-net on Cifar10 dataset. Sample quality measured by FID.

order of magnitude more weights. This makes a large comparison more expensive, and also risks overfitting on a small dataset like Cifar10. However, for completeness, and to verify the fairness of our comparison, we also report the FID that we were able to obtain with MMD-GAN on this fully-convolutional architecture in Table 2. Compared to our experiments using MMD-GAN to train the DCGAN architecture, the performance of MMD-GAN with the fully convolutional architecture remains unchanged for the larger CelebA dataset. On Cifar10, not surprisingly, the larger fully convolutional architecture performs slightly better than the DCGAN architecture trained using MMD-GAN. The difference in FID between the two different architectures is relatively small, justifying our decision to compare the generative training methods on the DCGAN architecture.

## 4.2 SENSITIVITY TO HYPERPARAMETERS AND EFFECT OF THE CRITIC DIMENSIONALITY

GAN training can be sensitive to the learning rate and the batch size used for training (Lucic et al., 2017). We examine the effect of learning rates and batch sizes on the performance of all three methods. Figure 5a compares the performance as a function of the learning rates. We see that GRAM-nets are much less sensitive to the learning rate than MMD-GAN, and are about as robust to changes in the learning rate as a vanilla GAN. MMD-GAN seems to be especially sensitive to this hyperparameter. We suggest that this might be the case since the critic in MMD-GAN is restricted to the set of $k$-Lipschitz continuous functions using gradient clipping, and hence needs lower learning rates. Similarly, Figure 5b shows the effect of the batch size on three models. We notice that all models are slightly sensitive to the batch size, and lower batch size is in general better for all methods.

We examine how changing the dimensionality $K$ of the critic affects the performance of our method. We use the Cifar10 data. Results are shown in Figure 5c. Interestingly, we find that there are two regimes: the output dimensionality steadily improves the FID until $K = 1000$, but larger values sharply degrade performance. This agrees with the intuition in Section 3.1 that dimensionality reduction is especially useful for an "intrinsically low dimensional" distribution. For more inspections of the stability of MMD-GANs, see Appendix C.

## 5 SUMMARY

We propose a new algorithm for training MMD-nets. While MMD-nets in their original formulation fail to generate high dimensional images in good quality, their performance can be greatly improved by training them under a low dimensional mapping. Unlike the alternative adversarial (MMD-GAN) for learning such a mapping, our training method, GRAM, learns this mapping while avoiding the saddle-point optimization by being trained to match density ratios of the input and the projected pair of distributions. This leads to a sizable improvement in stability and generative quality, that is at par with or better than adversarial generative methods.

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

## A   DERIVATION DETAILS

The derivation from (4) - (6) relies on the law of unconscious statistician (LOTUS), and therefore no Jacobian correction is needed. Here we show how we applied LOTUS in the derivation, with more detailed steps.

Let $\bar{r}(x) = \frac{\bar{p}(x)}{\bar{q}(x)}$. Expanding (4) we have,

$$
\begin{aligned}
D^*(\theta) &= \int q_x(x) \left( \frac{p_x(x)}{q_x(x)} \right)^2 dx - 2 \int q_x(x) \frac{p_x(x)}{q_x(x)} \frac{\bar{p}(f_\theta(x))}{\bar{q}(f_\theta(x))} dx + \int q_x(x) \left( \frac{\bar{p}(f_\theta(x))}{\bar{q}(f_\theta(x))} \right)^2 dx \\
&= \int p_x(x) \frac{p_x(x)}{q_x(x)} dx - 2 \int p_x(x) \frac{\bar{p}(f_\theta(x))}{\bar{q}(f_\theta(x))} dx + \int q_x(x) \left( \frac{\bar{p}(f_\theta(x))}{\bar{q}(f_\theta(x))} \right)^2 dx \\
&= \int p_x(x) \frac{p_x(x)}{q_x(x)} dx - 2 \int p_x(x) \bar{r}(f_\theta(x)) dx + \int q_x(x) \bar{r}^2(f_\theta(x)) dx
\end{aligned}
$$

We obtain the second line by canceling and rearranging terms and the third line, by plugging in $\bar{r}(x)$.

Applying Theorem 3.6.1 from Bogachev (2007) (pg. 190) on the third term, $\int q_x(x) \bar{r}^2(f_\theta(x))$, with $g = \bar{r}^2$ and $f = f_\theta$, we get

$$
\int q_x(x) \bar{r}^2(f_\theta(x)) dx = \int \bar{q}(f_\theta(x)) \bar{r}^2(f_\theta(x)) df_\theta(x).
$$

Similarly, for the second term, with $g = \bar{r}$ and $f = f_\theta$, we obtain

$$
2 \int p_x(x) \bar{r}(f_\theta(x)) dx = 2 \int \bar{p}(f_\theta(x)) \bar{r}(f_\theta(x)) df_\theta(x).
$$

Thus,

$$
D^*(\theta) = \int p_x(x) \frac{p_x(x)}{q_x(x)} dx - 2 \int \bar{p}(f_\theta(x)) \bar{r}(f_\theta(x)) df_\theta(x) + \int \bar{q}(f_\theta(x)) \bar{r}^2(f_\theta(x)) df_\theta(x),
$$

which is the first line of (5).

Note that we use LOTUS "in reverse" of its usual application.

## B   EXPERIMENTAL DETAILS AND MORE RESULTS FOR SECTION 3.4

### B.1   EXPERIMENTAL DETAILS

Below list the hyper-parameters used in Section 3.4, except from those being varied in the experiments.

- Number of epochs: 2,000
- Noise distribution: Gaussian
- Activation between hidden layers: ReLU
- Batch normalisation: not used
- Batch size: 200
- Batch size for generated samples: 200
- GAN
  - Optimizer: ADAM
  - Learning rate: 1e-4

- – Momentum decay: 0.5
- – Critic architecture: 2-100-100-10
- MMD-net
  - – Optimizer: RMSprop
  - – Learning rate: 1e-3
  - – RBF kernel bandwidth: 1
- GRAM-net
  - – Optimizer: ADAM
  - – Learning rate: 1e-3
  - – Momentum decay: 0.5
  - – RBF kernel bandwidth: 1
  - – Critic architecture: 2-100-100-10

Note that we also provide the experimental details for MMD-nets where we will show the results in Appendix B.3.

All parameter settings are also available in the `examples/Hyper.toml` file of our submitted code. All parameters being varied are also available in the `examples/parallel_exps.jl` file of our submitted code.

## B.2 3D RING DATASET

We also conduct the same experiment as in Figure 1a with a 3-dimensional data space and a 2-dimensional projection space. We create a 3D ring dataset by augmenting the third dimension with Gaussian noise (0 mean and 0.1 standard deviation), which is then rotated by 60 degrees along the second axis. We repeat the experiments in Figure 1a using this dataset. The results are shown in Figure 6. Unlike the 2D ring example, the optimal choice of the projection function learned here is no

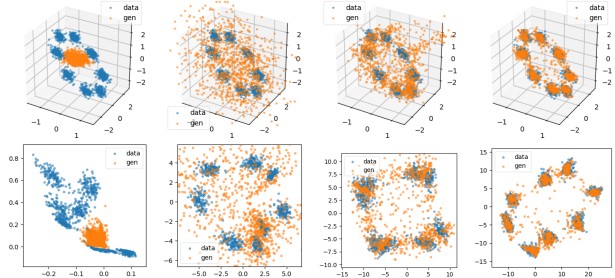

(a) Data and samples in the original (top) and projected space (bottom) during training; four plots are at iteration 0, 100, 1000 and 10, 000 respectively. Notice how the projected space separates $\bar{p}$ and $\bar{q}$.

Figure 6: Training results of GRAM-nets with projected dimension fixed to 2 on the 3D ring dataset.

longer the identity function. However, our method can stil, easily learn a low-dimensional manifold that tends to preserve the density ratio.

## B.3 STABILITY OF MMD-NETS AND MMD-GANS

In this section we repeat the same stability-related experiments that we conducted on GANs and GRAM-nets in Section 3.4, for MMD-nets and MMD-GANs. Results are shown in Figures 7a and 7b. One can see MMD-nets can capture all the modes, but the learned distribution is less separated (or sharp) compared to GRAM-nets. This is because the MMD is computed with a fixed kernel and in a fixed space, which can only distinguish distributions, subject to the set of kernels being used. Figure 7b shows the results on MMD-GAN modles that are trained for 4,000 epochs, each with 5

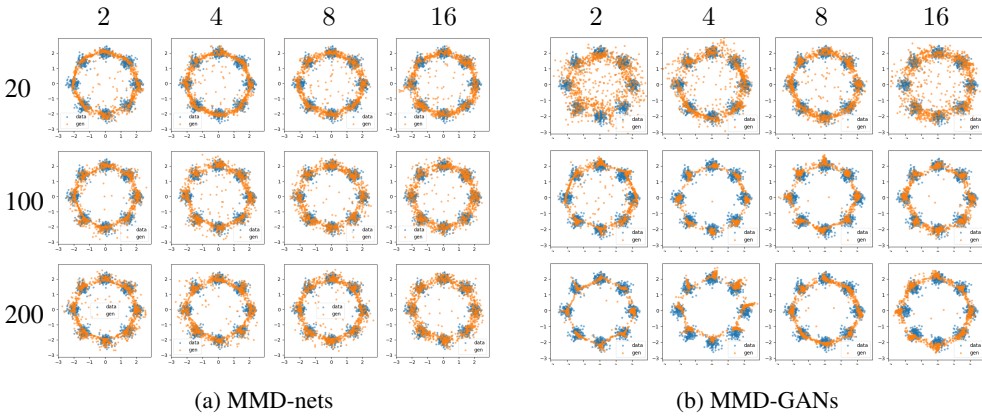

(a) MMD-nets           (b) MMD-GANs

Figure 7: Corresponding plots to Figure 2 for MMD-nets and MMD-GANs.

steps for the critic network and 1 step for the generative network.[6] Compared to Figure 2b, in which GRAM-nets only takes 2,000 training epochs with 1 joint step for both networks, even with longer training, the quality of MMD-GAN is visually worse than GRAM-nets in this synthetic example. Notice, how the generated samples of MMD-GAN are similar to the successful runs for GAN in Figure 2a: generated samples tend to be too concentrated around the mode of the individual clusters. Our method on the other hand, with shorter training time, is clearly able to recover all the 8 clusters along with their spreads.

Note that, unlike GRAM-nets which are robust to changing dimensionality of the projected spaces (2, 4, 8, 16), MMD-GAN training easily diverges for 2-dimensional projected spaces. As a result, for MMD-GAN we only show the results up to the iterations before the training diverges. It's likely that any change in the neural network size (output dimension) requires a different set of hyperparameters to make MMD-GAN converge. On the other hand, GRAM-nets are robust to these changes, out of the box.

### B.4 GRAM-NETS ON THE MNIST DATASET

In this section, we show the results of training GRAM-nets on the MNIST dataset. Figure 8a shows how the generated samples change during the phase of training. Figure 8b shows how the generator loss and the ratio matching objective are being optimized simultaneously. The generator used in this experiment is a multilayer perceptron (MLP) of size 100-600-600-800-784 with ReLU activations between hidden layers and sigmoid activation for the output layer; the noise distribution is a multivariate uniform distribution between $[-1, 1]$ with 100 dimensions. For the critic, we use a convolution architecture specified in Table 3. For the optimisation, we use ADAM with a learning rate of 1e-3, momentum decay of 0.5, and batch size of 200 for both data and generated samples. For the RBF kernels, we use bandwidths of $[0.1, 1, 10, 100]$

## C STABILITY OF MMD-GANS

In addition to Figure 7b, which evaluates the stability of MMD-GANs in terms of the projected dimension and the generator capacity, we perform qualitative evaluation by visualizing the projected space during training as well as the effect of stabilization techniques in this section.

### C.1 PROJECTED SPACE DURING TRAINING

We first perform the qualitative evaluation (same as Figure 1a and Figure 6) for MMD-GANs on the 2D and 3D ring datasets. The original space and the projected space during training are visualized

---

[6]When training MMD-GAN, both the auto-encoding loss and the feasible set reduction loss are used. Parameter clipping is done with -0.1 (lower) and 0.1 (upper). Learning rate is set to 5e-5 and other parameters are the same as GRAM-nets.

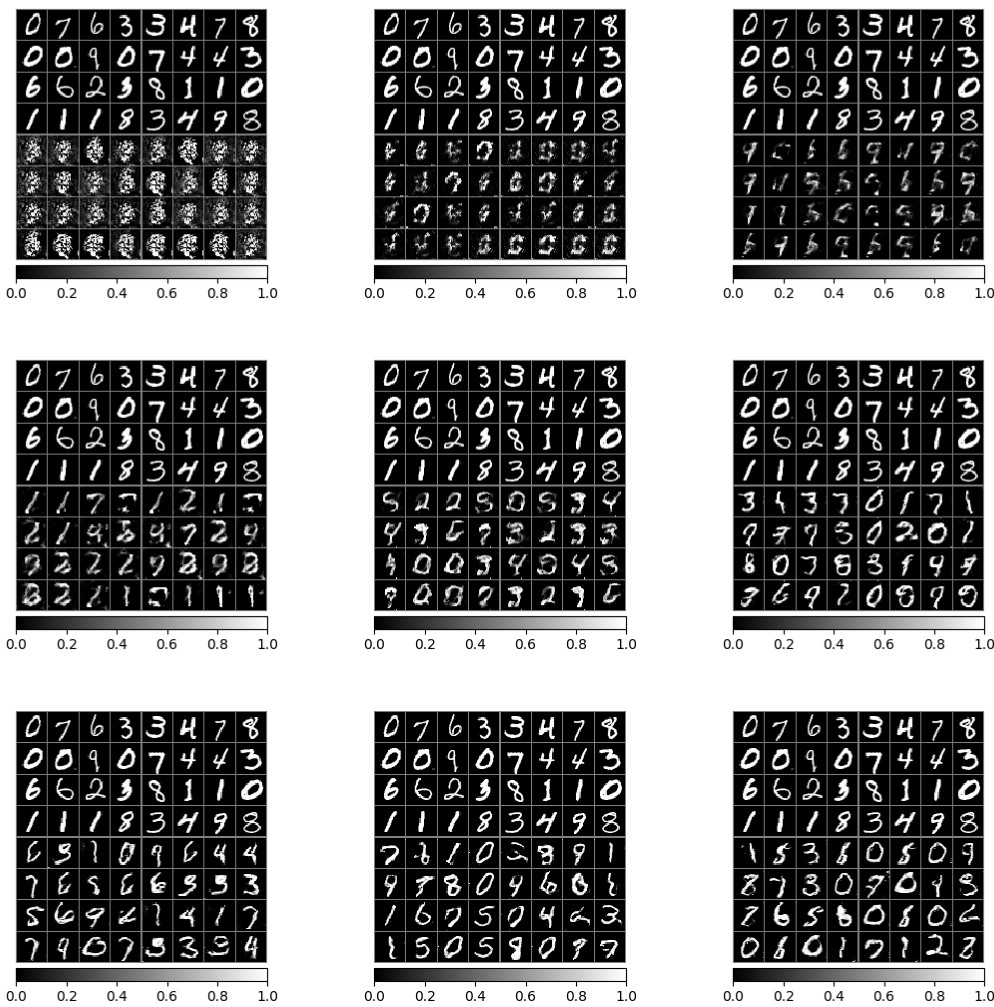

(a) Data and samples (top and bottom half in each plot) during training at iteration 100, 250, 500 (first row), at 750, 1,000, 2,000 (second row), and 3,000, 4,000 and 5,000 (last row). The orders for each row are from left to right.

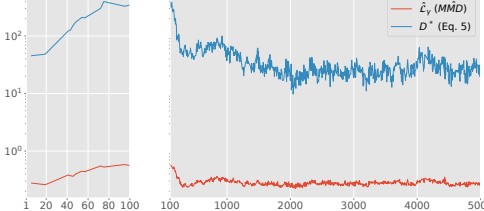

(b) Trace of $\hat{\mathcal{L}}_\gamma$ and $D^*$ (equation (5)) during training. The left plot is for iteration 1 to 100 and the right plot is for 100 to 5,000, with the same y-axes in the log scale.

Figure 8: Training results of GRAM-nets on the MNIST dataset.

in Figure 9 It is clear that the projected spaces are quite different between MMD-GAN and our method. Unlike GRAM-nets, in the projected space, the generated samples do not overlap with the data samples.

| | Op | Input | Output | Filter | Pooling | Padding |
|---|---|---|---|---|---|---|
| | Reshape | 784 | (-1,28,28,1) | - | - | - |
| | Conv2D + BatchNorm | 1 | 16 | 3 | 2 | 1 |
| | Conv2D + BatchNorm | 16 | 32 | 3 | 2 | 1 |
| | Conv2D + BatchNorm | 32 | 32 | 3 | 2 | 1 |
| | Reshape | (-1,3,3,32) | (-1,288) | - | - | - |
| | Linear | 288 | 100 | - | - | - |

Table 3: Critic architecture for MNIST. All BatchNorm are followed by a ReLU activation.

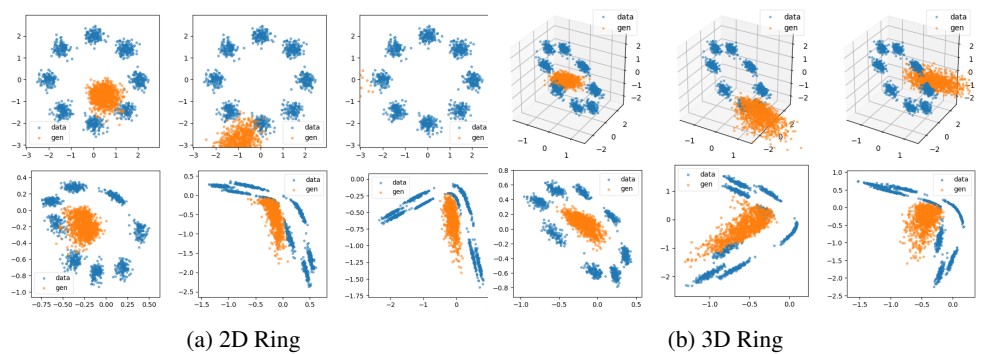

(a) 2D Ring            (b) 3D Ring

Figure 9: Training of MMD-GAN with projected dimension fixed to 2 *before diverging*. Data and samples in the original (top) and projected space (bottom) during training; four plots are at iteration $100$, $500$ and $1,000$ respectively. Notice how the projected space separates $\bar{p}$ and $\bar{q}$.

## C.2 STABILIZATION TECHNIQUES

We also evaluate the effect of the various stabilization techniques used for training, namely the autoencoder penalty (AE) and the feasible set reduction (FSR) techniques from (Li et al., 2017) on the Cifar10 data over two settings of the batch size. Table 4 shows the results. The performance of

Table 4: Performance of MMD-GAN (Inception scores; larger is better) for MMD-GAN with and without additional penalty terms: feasible set reduction (FSR) and the autoencoding loss (AE). The full MMD-GAN method is MMD+FSR+AE.

| Batch Size | MMD-GAN = MMD+FSR+AE | MMD+FSR | MMD+AE | MMD |
|---|---|---|---|---|
| 64 | $5.35 \pm 0.05$ | $5.40 \pm 0.04$ | $3.26 \pm 0.03$ | $3.51 \pm 0.03$ |
| 300 | $5.43 \pm 0.03$ | $5.15 \pm 0.06$ | $3.68 \pm 0.22$ | $3.87 \pm 0.03$ |

MMD-GAN training clearly relies heavily on FSR. This penalty not only stabilizes the critic but it can also provides additional learning signal to the generator. Because these penalties are important to the performance of MMD-GANs, it requires tuning several weighting parameters, which need to be set carefully for successful training.

We would like to re-emphasize the stability of GRAM-nets with respect to different settings of network sizes, noise dimension, projected dimension, learning rate, batch size without relying on regularization terms with additional hyperparameters.

## D  ARCHITECTURE

For the generator used in Section 4, we used the following DCGAN architecture,

| Op | Input | Output | Filter | Stride | Padding |
|---|---|---|---|---|---|
| Linear | 128 | 2048 | - | - | - |
| Reshape | 2048 | (-1,4,4,128) | - | - | - |
| Conv2D_transpose | 128 | 64 | 4 | 2 | SAME |
| Conv2D_transpose | 64 | 32 | 4 | 2 | SAME |
| Conv2D_transpose | 32 | 3 | 4 | 2 | SAME |

Table 5: DCGAN generator architecture for Cifar10.

We used two different architectures for the experiments on Cifar10 dataset. Table 6 shows the standard DCGAN discriminator that was used. Table 7 shows the architecture of the weaker DCGAN

| Op | Input | Output | Filter | Stride | Padding |
|---|---|---|---|---|---|
| Conv2D | 3 | 32 | 4 | 2 | SAME |
| Conv2D | 32 | 64 | 4 | 2 | SAME |
| Conv2D | 64 | 128 | 4 | 2 | SAME |
| Flatten | 128 | 2048 | - | - | - |
| Linear | 2048 | 128 | - | - | - |

Table 6: DCGAN discriminator architecture for Cifar10.

discriminator architecture that was also used for Cifar10 experiments. While leaky-ReLU was used

| Op | Input | Output | Filter | Stride | Padding |
|---|---|---|---|---|---|
| Conv2D | 3 | 32 | 4 | 2 | SAME |
| Conv2D | 32 | 32 | 4 | 2 | SAME |
| Conv2D | 32 | 64 | 4 | 2 | SAME |
| Flatten | 64 | 1024 | - | - | - |
| Linear | 1024 | 128 | - | - | - |

Table 7: Shallow DCGAN discriminator architecture.

as non-linearity in the discriminator, ReLU was used in the generator, except for the last layer, where it was tanh. Batchnorm was used in both the generator and the discriminator.

# E  SAMPLES

We show some of the randomly generated samples from our method on CIFAR10 and CelebA in Figure 10.

# F  INCEPTION SCORE

Inception score (IS) is another evaluation metric for quantifying the sample quality in GANs. Compared FID, IS is not very robust to noise and cannot account for mode dropping. In addition to the FID scores that we provide in the paper, here we also report IS for all the methods on CIFAR10 for completeness since the MMD-GAN paper used it as their evaluation criteria.

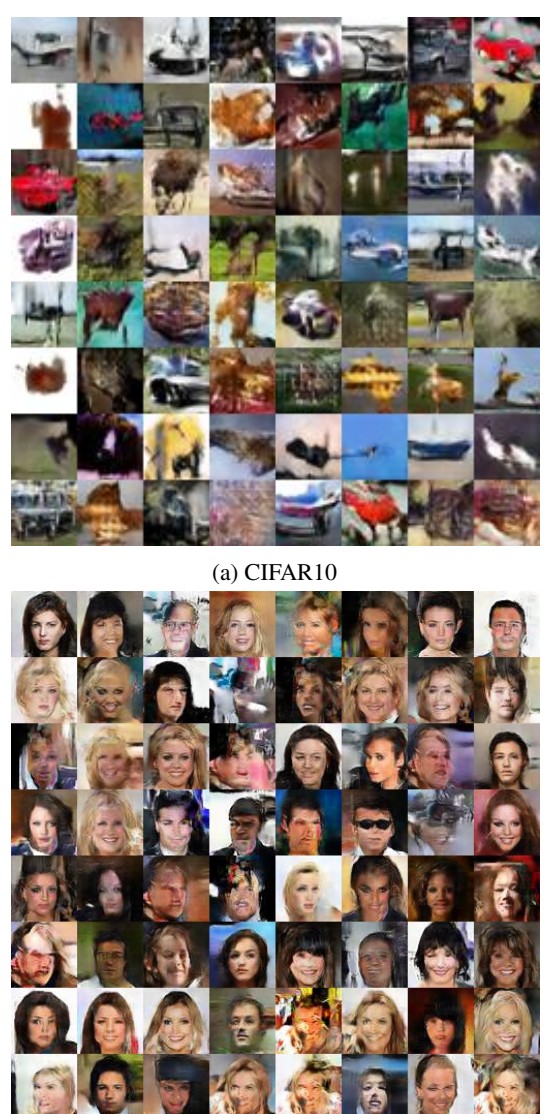

(a) CIFAR10

(b) CelebA

Figure 10: Random Samples from a randomly selected epoch (>100).

Table 8: Inception Scores for MMD-GAN, GAN, GRAM-net and MMD-nets on CIFAR10 for three random initializations.

|  | **MMD-GAN** | **GAN** | **GRAM-net** |
|---|---|---|---|
| **Inception Score** | $5.35 \pm 0.12$ | $5.17 \pm 0.13$ | $\mathbf{5.73 \pm 0.10}$ |
|  | $5.21 \pm 0.14$ | $4.94 \pm 0.15$ | $\mathbf{5.44 \pm 0.12}$ |
|  | $5.31 \pm 0.10$ | $5.27 \pm 0.05$ | $\mathbf{5.45 \pm 0.18}$ |

