# OpenReview forum: "Generative Ratio Matching Networks"
_ICLR.cc/2020/Conference — Accept (Poster)_

### Official Review · AnonReviewer2 · 2019-10-16
**Official Blind Review #2**

**Rating:** 6

**Review:**

The proposed approach amounts to a non-adversarial way to update the parameters theta of a kernel K_theta(x, y) = k(f_theta(x), f_theta(y)) in an MMD-net, rather than adversarially as in MMD-GANs. Avoiding the saddle point problem clearly gives a more stable optimization problem, as shown in Section 3.4, which is very nice.


Theory/motivation:

The motivation based on density ratio estimation, however, very much assumes that these densities exist. As convincingly argued by Arjovsky and Bottou (ICLR 17, "Towards principled methods for training generative adversarial networks"), the densities p_x, q_x of (4) typically do not exist (w.r.t. Lebesgue measure), and moreover the support of the distributions are typically almost disjoint. Algorithmically, this doesn't seem to be a major issue for your approach: the estimator (8) corresponds to some kind of smoothing by the kernel. But it would be nice from a motivation point of view to have a better understanding of:

- What assumptions exactly does (8) make about the existence of densities? For example, in the case of a Gaussian kernel, does it correspond to the distance between distributions convolved with Gaussian noises with a smaller Gaussian kernel? Does the algorithm "make sense" when distributions don't exist?

- Can we say something about when a density exists for the dimensionality-reduced data f_theta(x) even when it doesn't exist for the data x?

- Relatedly, the dimension-reduced Pearson divergence is clearly somehow different from the plain Pearson divergence. Can we understand a little bit more how it's different? How does the choice of architecture of f_theta affect the distance metric?

- Is it the case, as in Arjovsky et al. (2017) or in Arbel et al. (NeurIPS 2018, "On gradient regularizers for MMD GANs"), that the overall loss function with an optimal f_theta is continuous with respect to the GAN distribution?

Certainly not all of these theoretical questions require answers for an initial ICLR paper, but they would be illuminating for understanding the method.


Motivating experiments:

- In Figure 2, it seems that except in the first image (h=2) the mapping f_theta becomes essentially the identity. Is this expected to be generally true in your 2d cases, because the density ratio can already be well-represented there? A more interesting example for this aspect of the model might involve higher-dimensional distributions with some low-dimensional manifold structure; would f_theta pick up some reasonable representation of the manifold in those cases?

- Can you plot the density ratio estimate in a simple 2d example like this? How does it look visually? Is a good estimate necessary for good performance of your generative model?


Experiments:

- As shown by Binkowski et al., it's extremely important to specify the sample size for FID scores, which you don't do.

- That said, the FID scores of the MMD GAN models of Binkowski et al using DCGAN discriminators (which replace the FSR and AE penalties of Li et al. with a single gradient penalty) found substantially smaller penalties for performance of MMD-GANs with using a smaller discriminator (1/4 as many hidden units rather than 1/2 as you use) than you did. They also reported substantially better numbers on CelebA than you did. If not too difficult code-wise, it might be worth trying either their code or that of the followup Arbel et al. (citation above) to compare to your approach.

- The results of, for example, Mescheder et al. (ICML-18, "Which Training Methods for GANs do actually Converge?") and Arbel et al. (NeurIPS-18, "On gradient regularizers for MMD GANs") seem to be much better visually with comparably-sized models than the results of Figure 8 (and of course StyleGAN/etc are *far* better but with drastically larger models). It would be good to compare empirically to some more recent models than original MMD GANs (whose empirical results were substantially improved with only minor tweaks by Binkowski et al. mere months later) or DCGAN, which are fairly old approaches in this space.

- Two more extremely relevant methods it would be worth comparing to or at least mentioning:

   - Ravuri et al. (ICML 2018, "Learning Implicit Generative Models with the Method of Learned Moments") can be interpreted as learning f_theta for an MMD-net as you do by taking the gradients of an auxiliary classification network. Although it is still somewhat "adversarial," they need to update the kernel function extremely rarely (every 2,000 generator update steps) and the optimization seems overall much more stable.

   - dos Santos et al. (https://arxiv.org/abs/1904.02762 "Learning Implicit Generative Models by Matching Perceptual Features") effectively use an MMD-net (with some slight tweaks) with a *fixed* kernel. Though results aren't quite as good, it's worth at least mentioning.


Minor typos/etc:

- Top of page 3: it is sufficient to choose F a unit ball in an RKHS with a _characteristic_ kernel.
- Page 4: "this issue in next."
- Figure 1 caption: (both) should probably be (bottom).


Overall initial thoughts:

This seems like a nice alternative to adversarial methods, but it does not compare to more recent (last 2 years) models in this space or very thoroughly establish the applicability of its motivation. I think it's worthy of a weak accept as is, but could be much more convinced with some additional work.

**Experience Assessment:**

I have published in this field for several years.

**Review Assessment: Checking Correctness Of Derivations And Theory:**

I assessed the sensibility of the derivations and theory.

**Review Assessment: Checking Correctness Of Experiments:**

I assessed the sensibility of the experiments.

**Review Assessment: Thoroughness In Paper Reading:**

I read the paper thoroughly.

---

> ### Author Response · Authors · 2019-11-11
> **We thank reviewer 2 for their valuable feedback. We have re-run the experiments that you pointed out and have updated the draft.**
>
> 1. Theory/Motivation: We do not assume that the data has a density with respect to Lebesgue measure, and indeed we expect in practice that the support of the data distribution is a low dimensional manifold. The reason for this is that MMD (like all the other IPMs, Integral probability metrics, with weaker topological assumptions) is well defined even when the “distributions don’t exist”, i.e are supported only over some underlying low-dimensional manifold of the data space (Arjovsky et al., 2017). That said, in our work, we additionally project the densities back to the same low dimensional manifold as the noise. We further provide empirical results on how changing the dimensionality of this manifold affects the generation in Figure 5(c).
>
> 2. On Pearson Divergence: This divergence uses an MMD-based ratio estimator in its own estimation. Therefore, it depends on the efficiency of the MMD-based density ratio estimator. Strictly speaking, it is not PD, as the ratio comes from MMD which has different topological assumption than f-divergences. As a result, this estimator is well defined even when the two densities are not absolutely continuous with respect to each other. The architecture of f_theta (except for the extreme cases) should not have an impact on the properties of this objective as long as the change of variable that f_theta implies is well defined.
>
> 3. On Figure 2: Can the reviewer please clarify how did they infer $f_\theta$ from figure 2, as it does not show the projected space? Additionally, Figure 1 (which shows the projected space) confirms that $f_\theta$ is not learning an identity mapping. We have added a new experiment, as you asked, where the data lives on a higher/different dimensional space than the projected space in Appendix F (Figure 10) to support our claim. Please note that training diverges if we attempt the same experiment for MMD-GAN (i.e. projecting to 2D space) as shown in the same appendix in Figure 11.
>
> 4. FID: We used MMD-GAN’s author’s original code and used 30000 samples for evaluating FID.
>
> 5. Small Critic Evaluation: Binkowski et al (2018) used several learning rate heuristics. For example, they adaptively decrease the learning rate using 3-sample test. Additionally, they employ several regulations (on both, weights and activations) to allow the network to train without diverging. We chose not to apply any additional tricks (more imp. no architectural specific regularisations) on any of the methods in order to fairly evaluate them all on exactly the same settings. As a result of this, MMD-GAN training (with the smaller critic) frequently diverged, leading to the high FID that we reported). So far we have been able to run their original code on the Cifar10 dataset with the default settings for 24 hours. After 150000 iterations it achieved an FID of 72.09 and an inception score of 5.02.
> Their celebA preprocessing (scaling/res) is different than the usual (64x64) and they use quite large networks (resnet or 5 layer DCGAN). As a result, unlike most GAN methods it requires at least 2 GPU (as per original code documentation) to run this experiment and leads to the higher reported FID. In light of these differences, we’d be happy to attempt full comparison if it would be useful.
> It is also common to centre-crop the faces in most recent work that removes the background and improves FID. We do not do this and therefore our samples look different in quality than those generated by newer models.
>
> 6. References: Thanks for the references. We will add them to our revised version. We did not compare ours to Ravuri et al 2018 since unlike our method, their method relies on labels. We would like to point out that GRAMnet can be used in a class-conditional setup as well.

---

> > ### Comment · AnonReviewer2 · 2019-11-11
> > **A few comments**
> >
> > Thanks for your responses. A few notes:
> >
> > 1. Certainly IPMs are well-defined in this situation (and of course I made a mistake in writing that the "distributions" don't exist above). But the quantity $r(x) = p(x) / q(x)$ which you try to estimate in (3) and (7) does not; even if we fiddle with things appropriately so that it does exist, it will often be infinite where $q(x)$ has no support and $p(x)$ does. It would be nice to understand this better.
> >
> > 2. Your derivation around (5) depends on $f_\theta$ being injective. Since it won't be exactly, the choice of $f_\theta$ should matter to your final objective. Figure 5c also shows that it clearly matters to at least the combination of objective + the optimization procedure, though of course these things are difficult to tease apart in deep learning settings.
> >
> > 3. Sorry, I meant Figure 1, specifically 1a's bottom row; all but the leftmost plot appear to be essentially (though not exactly) the identity function here. The new Figure 9 does help resolve the question I had; clearly in this case it's doing a reasonable thing. It might be nice to understand what it's doing in e.g. an image case, but that's of course more difficult to visualize. Incidentally, in the [hopeful] final version you should integrate Appendix F into the rest of the appendices, not single it out as "here are the things we added to appease the reviewers"!
> >
> > 4. Thanks; this is a reasonable choice but the number should be in the paper.
> >
> > 5. I don't think a full comparison to Binkowski et al.'s code is necessary, but it would be helpful to explicitly say that the heuristics of that paper seem to help MMD GANs and the model does worse without them.
> >
> > 6. Ravuri et al. don't actually use class labels from the dataset; their "moment network" is trained as a standard discriminator between training and generator samples. You don't necessarily need to add experiments comparing to this approach, though it would be nice, but it's worth at least mentioning.

---

> > > ### Author Response · Authors · 2019-11-12
> > > **Thanks for your valuable comments and feedback once again, we will incorporate all the suggested changes and references.**
> > >
> > > 1. True density ratio for distributions without overlapping supports may evaluate to zero (in eqs. 3-7 and not infinity as the expectation in these eqs are always with respect to the denominator density) for some subset of points in a fixed design approach. This is why we estimate the ratio in the projected low-dimensional spaces using the MMD criterion, that alleviates this issue by convolving the distributions with a gaussian kernel.
> > >
> > > 2. Unlike MMD-GAN that needs to ensure that the composite of the learned kernel and the rbf kernel is still a valid kernel, injectivity is not required in the derivation around (5). Our method needs the function $f_\theta$ to be such that $\int_{-\infty}^\infty |f_\theta(x)| p(x) < \infty$ in order to invoke LOTUS. Also, as you pointed out the dimensionality of the projected space affects the outcome and there is a trade-off here. If the projected dimension is too small, (4) can not be minimised to a small enough quantity to ensure the density ratio is well-preserved. If the projected dimension is too large, both the density ratio estimation and MMD loss would not perform well.
> > >
> > > 3. We did so simply to clearly point out to the reviewers where to find the changed parts and new experiments in the paper. We will most certainly change it post review.

---

> > > > ### Comment · AnonReviewer2 · 2019-11-12
> > > > **Change of variables**
> > > >
> > > > 1. True; good point.
> > > >
> > > > 2. MMD-GAN injectivity is a different issue: it is always a valid kernel, the only reason for the transformation to be injective is to ensure the kernel is characteristic. As demonstrated by Binkowski et al. / Arbel et al., this isn't actually necessary; see e.g. the latter's Proposition 1 for a heuristic argument.
> > > >
> > > > On further thought about your case: you're not actually quite invoking LOTUS, as far as I can see, but rather doing a vector-to-vector change of variables https://en.wikipedia.org/wiki/Probability_density_function#Vector_to_vector - which requires either that $f_\theta$ be bijective, and in retrospect I haven't checked that the various Jacobian terms cancel out appropriately, or you need to correct for the points which are collapsed together appropriately. This is possibly what R1 was referring to as a mistake in the derivation, though they haven't returned to clarify that.

---

> > > > > ### Author Response · Authors · 2019-11-13
> > > > > **Clarification**
> > > > >
> > > > > We respectfully disagree with the reviewer. The derivation from (4) - (6) in our paper does indeed rely on LOTUS, and no Jacobian correction is needed, therefore. Here is how we apply LOTUS with more detailed steps.
> > > > > Define $\bar{r}(x) = \frac{\bar{p}(x)}{\bar{q}(x)}$ and start with the expanded version of (4)
> > > > >
> > > > > $$D^*(\theta) = \int q_x(x) \left( \frac{p_x(x)}{q_x(x)} \right)^2 dx - 2 \int q_x(x) \frac{p_x(x)}{q_x(x)} \frac{\bar{p}(f_\theta(x))}{\bar{q}(f_\theta(x))} dx + \int q_x(x) \left( \frac{\bar{p}(f_\theta(x))}{\bar{q}(f_\theta(x))} \right)^2 dx $$
> > > > > $$= \int p_x(x) \frac{p_x(x)}{q_x(x)} dx - 2 \int p_x(x) \frac{\bar{p}(f_\theta(x))}{\bar{q}(f_\theta(x))} dx + \int q_x(x) \left( \frac{\bar{p}(f_\theta(x))}{\bar{q}(f_\theta(x))} \right)^2 dx$$
> > > > > $$= \int p_x(x) \frac{p_x(x)}{q_x(x)} dx - 2 \int p_x(x) \bar{r}(f_\theta(x)) dx + \int q_x(x) \bar{r}^2(f_\theta(x)) dx$$
> > > > >
> > > > > We obtain the second line by cancelling and rearranging terms. We obtain the third line by plugging in $\bar{r}(x)$ (defined in the beginning).
> > > > >
> > > > > First consider the third term $\int q_x(x) \bar{r}^2(f_\theta(x))$. We can apply Theorem 3.6.1 from Bogachev, V.I., 2007; page 190 of https://tinyurl.com/wlsavp3 , with $g = \bar{r}^2$ and $f = f_\theta$, which gives us
> > > > >
> > > > > $$\int q_x(x) \bar{r}^2(f_\theta(x)) dx = \int \bar{q}(f_\theta(x)) \bar{r}^2(f_\theta(x)) df_\theta(x)$$
> > > > >
> > > > > Similarly, for the second term, with $g=\bar{r}$ and $f = f_\theta$, we obtain
> > > > >
> > > > > $$2 \int p_x(x) \bar{r}(f_\theta(x)) dx = 2 \int \bar{p}(f_\theta(x)) \bar{r}(f_\theta(x)) df_\theta(x)$$
> > > > >
> > > > > Thus we have
> > > > >
> > > > > $$D^*(\theta) = \int p_x(x) \frac{p_x(x)}{q_x(x)} dx - 2 \int \bar{p}(f_\theta(x)) \bar{r}(f_\theta(x)) df_\theta(x) + \int \bar{q}(f_\theta(x)) \bar{r}^2(f_\theta(x)) df_\theta(x),$$
> > > > >
> > > > > which is the first line of (5). Note that we use LOTUS "in reverse" of its usual application.
> > > > >
> > > > > More simply, using the vector-to-scalar change of variable from the link you provided (https://en.wikipedia.org/wiki/Probability_density_function#Vector_to_scalar), one can also show the same. In (4), the (whole) integrand is in fact, a vector-to-scalar function (and not vector-to-vector function). Let's denote it as $g = \bar{r}^2(f_\theta(x))$ with density $p_g$. Let $y=f_\theta(x)$ with density $p_y$ then using LOTUS we have,
> > > > >
> > > > > $$\int g p_g(g) dg = \int r^2(y) p_y(y) dy = \int r^2(f_\theta(x)) p_x(x) dx.$$
> > > > >
> > > > > Applying the above directly to (4) we transition to (5), i.e. we apply LOTUS at two different levels, once wrt to the density of $f_\theta$ and then wrt to the density of $x$.
> > > > >
> > > > > Further, we have also added a simulation-based proof of the same in appendix G.

---

> > > > > > ### Comment · AnonReviewer2 · 2019-11-15
> > > > > > **Thanks!**
> > > > > >
> > > > > > Thank you for the very clear refutation of my doubts. I was indeed confused by the LOTUS being in "reverse," but I'm happy now. :)
> > > > > >
> > > > > > It might be helpful for the final version of the paper to include this more detailed derivation in the appendix.

---

### Official Review · AnonReviewer3 · 2019-10-19
**Official Blind Review #3**

**Rating:** 6

**Review:**

In this paper, authors propose a new generative adversarial networks based on density-ratio estimation. More specifically, the squared-loss between the true ratio function and the model  (a.k.a., Pearson divergence) is employed to train model. Through experiments, authors demonstrate that the proposed density-ratio based approach compares favorably with existing algorithm.

Experimentally, the proposed model performs pretty nicely. However, the derivation is not certain and some important density-ratio based GAN is missing.

Detailed comments:
1. Some important density-ratio based GAN methods are missing. For instance, https://arxiv.org/abs/1610.02920 https://arxiv.org/abs/1610.03483
2. Some derivation of density-ratio estimation is not clear. To the best of my knowledge, the ratio model is obtained by minimizing the distance between true ratio function and its ratio model (Something similar to (4)). But, in this paper, the authors describe the ratio is obtained by using MMD density ratio estimator. Could you explain the derivation in detail?
3. I guess, (8) is derived in similar way of  http://www.jmlr.org/papers/volume10/kanamori09a/kanamori09a.pdf In this case, (8) is not correct.
4. The ratio is defined by using kernel. Thus, I guess the kernel model is employed. However, this information is missing.

**Experience Assessment:**

I have published in this field for several years.

**Review Assessment: Checking Correctness Of Derivations And Theory:**

I assessed the sensibility of the derivations and theory.

**Review Assessment: Checking Correctness Of Experiments:**

I assessed the sensibility of the experiments.

**Review Assessment: Thoroughness In Paper Reading:**

I read the paper at least twice and used my best judgement in assessing the paper.

---

> ### Author Response · Authors · 2019-11-11
> **We thank reviewer 3 for their valuable feedback. We have updated the draft with their suggestions.**
>
> 1. Thanks for the references - we will add them to our paper. These methods (1,2) are different from ours.
> (1) They use direct density ratio estimators to compute the divergence for training the generator. In our method, the generator is trained via the MMD loss which in turn is used to create a density ratio estimator (DRE). This DRE is used to train the critic via a novel ratio matching objective.
> (2)  We will add this to our references but we would like to point out that the use of classifier for DRE dates back to work done well before this paper (as the paper points out).
>
> 2. DRE Derivation: No, we do not follow Kanamori et al to derive (8); in particular, we do not use a model of the density ratio. Instead, our DRE follows fixed-design setup, that is, we are not trying to estimate the density ratio for all possible points, but only at a fixed set of points, namely, the points ${ f(x^q_1), \ldots, f(x^q_M) }$, which are computed in line 3 of Algorithm 1. We obtain the DRE (8) as the minimum of (3). For the detailed derivation of this, we would like to refer the reviewer to Chapter 3 of the book we referenced (Sugiyama et al., 2012; [https://www.cambridge.org/core/books/density-ratio-estimation-in-machine-learning/BCBEA6AEAADD66569B1E85DDDEAA7648](https://www.cambridge.org/core/books/density-ratio-estimation-in-machine-learning/BCBEA6AEAADD66569B1E85DDDEAA7648)). The MMD ratio estimator we used is provided in Section 3.3.1.
>
> 3. Kernel Information: We use Gaussian / RBF kernels. This information is provided in Algorithm 1. We also provide the exact bandwidths we used for Section 3.4 in the appendix. For all the other experiments, we used bandwidths [1, 2, 4, 8, 16]. The same set of kernels are used to train the generator and estimate the density ratios, as it is described in Line 4 of Algorithm 1.

---

> > ### Comment · AnonReviewer3 · 2019-11-15
> > **Thanks for the response**
> >
> > > 2. DRE Derivation: No, we do not follow Kanamori et al to derive (8); in particular, we do not use a model of the density ratio.
> >
> > Okay. Now I understood. Actually, authors must refer to the original paper of MMD based density-ratio estimation in addition to the book.
> >
> > http://papers.nips.cc/paper/3075-correcting-sample-selection-bias-by-unlabeled-data.pdf
> >
> > Then, I am happy to raise the score to 6.

---

### Official Review · AnonReviewer1 · 2019-10-22
**Official Blind Review #1**

**Rating:** 6

**Review:**

This paper proposes GRAM-nets using MMD as the critic to train GANs. Similar to MMD-GAN, the MMD is computed on a projected lower dimensional space to prevent the kernel struggle in the high dimensional observed space. On the other hand, contrary to MMD-GAN, GRAM-nets trains the projection f_{\theta} trying to matching the density ratio of p/q between the observed and the latent space. The proposed density ratio matching criterion avoids the adversarial training in MMD-GAN, thus can potentially fix the two-player optimization problem. The paper shows improved FID scores and nice-looking CIFAR10 generations.

Strengths,
1, Matching density ratios is a novel and interesting idea. If the data lives in a lower dimensional subspace, matching density ratio could probably reveal the subspace. Compared to adversarially training f_theta, the proposed approach could lead to more stable training potentially.
2, By manipulating E (px/qx - pz/qz)^2, they avoid estimating the high-dimensional px/qx and only estimate pz/qz.


Weakness,
1, The paper needs to be more careful with mathematical expressions. 1) Eq(2) should be MMD^2, instead of MMD. 2) Eq(5) and Eq(6) are wrong, although the used Eq(7) becomes true again. In Eq(5), Eq(6), the integration should be over f(x) instead of x, that is to say $\int ..... df(x)$.
2, It is unclear why one needs the regularization in Eq(9). In fact, the major problem of the density ratio estimator lies in that r(x) might be negative, so a clipping might be useful.
3, The major contribution of GRAM-nets lies in removing the adversarial training in MMD-GAN. Therefore, more empirical comparisons should be made with MMD-GAN. For example, how MMD-GAN evolves in Figure 2 is necessary.
4, In real image generation tasks, it is beneficial to show the stability of training GRAM-nets, compared with GAN, MMD-GAN as well as WGAN-GP; And Inception scores should also be reported for better validating the effectiveness of the proposed method.

**Experience Assessment:**

I have read many papers in this area.

**Review Assessment: Checking Correctness Of Derivations And Theory:**

I carefully checked the derivations and theory.

**Review Assessment: Checking Correctness Of Experiments:**

I assessed the sensibility of the experiments.

**Review Assessment: Thoroughness In Paper Reading:**

I read the paper thoroughly.

---

> ### Author Response · Authors · 2019-11-11
> **We thank reviewer 1 for their valuable feedback and suggestions. We have tried to incorporate these in the revised submission. .**
>
> 1. On integration measure: Thanks for correctly pointing out, indeed, the integrals are defined with respect to the measure $f_\theta(x)$. (7) becomes correct again after invoking LOTUS (law of unconscious statistician). We will fix these typos. We would also like to clarify if this is what you referred to as the 'fundamental error'? If not, could you please point it out.
>
> 2. Clipping DRE: Our motivation to add the non-negativity regularization term is to encourage f_theta to produce a reduced space on which the density ratio estimator tends to produce positive values, which is trying to alleviate the problem that you pointed out. In fact, we tried your suggestion on the synthetic datasets and MNIST and found that clipping the estimator works very robustly. Thanks for this great suggestion!
>
> 3. Synthetic data experiment for MMD-GAN: We ran the experiments from Figure 2 for MMD-GANs, which are provided in Figure 10 (Appendix F). Note that these plots take 4,000 epochs, each with 5 steps for the critic network and 1 step for the generative network, while Figure 2b for GRAM-nets only takes 2,000 epochs with a single joint step for both the networks. Even with longer training, the quality of MMD-GAN is visually worse than GRAM-nets in this synthetic example. Notice, how the generated samples of MMD-GAN are similar to the successful runs for GAN in Figure 2a: generated samples tend to be too concentrated around the mode of the individual clusters. Our method on the other hand, with shorter training time, is clearly able to recover all the 8 clusters along with their spread
>
> 4. On comparison with WGAN-GP: Wasserstein distance and MMD are both instances of integral probability metrics and have been compared to each other before in great details by Binkowski et al, (2018). They were able to show that MMD-GAN is better (stable, efficient with higher IS) than WGAN-GP, therefore we chose to compare to MMD-GAN in this work given its direct relevance. Figure 5 shows the stability of GRAM-nets v.s. MMD-GANs. We have added inception scores for MMD-GANs, GANs and GRAM-nets on CIFAR10 in Appendix E, as per your suggestion.

---

> > ### Comment · AnonReviewer1 · 2019-11-13
> > **Thanks for the clarifications**
> >
> > Thanks for your rebuttal.
> >
> > 1) It is the "fundamental error" I referred to. But as it was only a typo in the final, it is not fundamental.
> > 2) The regularization makes more sense with your explanation.
> > 3,4) The new experiments look pretty good, the proposed method seems indeed more stable than MMD-GAN. A minor concern is that for the inception score with DCGAN, there are actually better results than yours https://openreview.net/pdf?id=r1laEnA5Ym , so it is probably worthwhile to check the difference.
> >
> > Overall, I think this paper presents an interesting method and the experiments are convincing. It should be accepted.

---

### Decision · Program_Chairs · 2019-12-19

**Decision:**

Accept (Poster)

**Comment:**

The paper proposes a training method for generative adversarial network that avoids solving a zero-sum game between the generator and the critic, hence leading to more stable optimization problems. It is similar to MMD-GAN, in which MMD is computed on a projected low-dim space, but the projection is trained to match the density ratio between the observed and the latent space.
The reviewers raised several questions. Most of them have been addressed after several rounds of discussions. Overall, they are all positive about this paper, so I recommend acceptance. I encourage the authors to incorporate those discussions in their revised paper.